# A Pressure Sensing Device to Assist in Colonoscopic Procedures to Prevent Perforation—A Case Study

**DOI:** 10.3390/s24175711

**Published:** 2024-09-02

**Authors:** Se-Eun Kim, Young-jae Kang, Chang-ho Jung, Yongho Jeon, Yunho Jung, Moon Gu Lee

**Affiliations:** 1Department of Mechanical Engineering, Ajou University, 206 Worldcup-ro, Yeongtong-gu, Suwon-si 16499, Republic of Korea; tpzi08@ajou.ac.kr (S.-E.K.); youngjae.kang01@gmail.com (Y.-j.K.); did9594@gmail.com (C.-h.J.); princaps@ajou.ac.kr (Y.J.); 2Department of Internal Medicine, College of Medicine, Soonchunhyang University, 22 Soonchunhyang-ro, Sinchang-myeon, Asan-si 31538, Republic of Korea; yoonho7575@naver.com

**Keywords:** pressure sensor, colonic perforation, colonoscope, medical device

## Abstract

Colonoscopy has a limited field of view because it relies solely on a small camera attached to the end of the scope and a screen displayed on a monitor. Consequently, the quality and safety of diagnosis and treatment depend on the experience and skills of the gastroenterologist. When a novice attempts to insert the colonoscope during the procedure, excessive pressure can sometimes be applied to the colon wall. This pressure can cause a medical accident known as colonic perforation, which the physician should prevent. We propose an assisting device that senses the pressure applied to the colon wall, analyzes the risk of perforation, and warns the physician in real time. Flexible pressure sensors are attached to the surface of the colonoscope shaft. These sensors measure pressure signals during a colonoscopy procedure. A simple signal processor is used to collect and process the pressure signals. In the experiment, a colonoscope equipped with the proposed device was inserted into a simulated colon made from a colon extracted from a pig. The processed data were visually communicated to the gastroenterologist via displays and light-emitting diodes (LEDs). The device helps the physician continuously monitor and prevent excessive pressure on the colon wall. In this experiment, the device appropriately generated and delivered warnings to help the physicians prevent colonic perforation. In the future, the device is to be improved, and more experiments will be performed in live swine models or humans to confirm its efficacy and safety.

## 1. Introduction

Colonoscopy is a method of diagnosing and treating inflammation, tumors, and polyps in the colon by inserting a colonoscope through the anus into the large intestine and observing the bowel wall. During these procedures, abnormalities in the bowel may be detected, followed by an immediate biopsy [1]. Furthermore, polypectomy and/or hemostasis are possible if the gastroenterologist identifies a polyp and/or bleeding during the procedure. Although colonoscopy is a relatively safe procedure, it can cause serious complications such as bleeding, infectious disease, spleen damage, and perforation. In particular, bowel perforation should be avoided because morbidity and mortality increase if its diagnosis and treatment are delayed [2].

Colonic perforation usually occurs when excessive pressure or tension is applied to the colon wall after the considerable looping of the flexible shaft of the colonoscope. The loop is sometimes created when the gastroenterologist attempts to pass the scope through the sigmoid colon or the hepatic flexure. Gastroenterologists usually recommend applying manual pressure on the patient’s abdomen or changing the patient’s position to address this problem. They may also employ techniques such as right/left torque, shaft pulling, and accordion-like pleating. However, the success rate of reaching the cecum from the anus varies, primarily depending on the experience and skill level of the gastroenterologist, which poses a significant entry barrier for novices [3].

To address this problem, Evans et al. have proposed improving colonoscopic skills for inspection, lesion characterization, and lesion removal [4]. Levine et al. have conducted a study to standardize endoscopic skills by using kinematic data generated during the procedure from wearable sensors on the trainee’s arm [5]. Nevertheless, there are instances where novices do not recognize the creation of a loop and continue to push the scope shaft, causing excessive pressure and tension on the colon wall, which can lead to colonic perforation. This may result in medical disputes between patients and medical staff.

Biomedical engineers have also made efforts to solve problems that arise during endoscopic diagnosis and treatment. Gerald et al. conducted a study to detect bleeding by attaching a small sensor to the surface of the endoscope shaft [6]. Javazm et al. attached a balloon filled with air to the outside of the endoscope and diagnosed polyps by changes in pressure [7]. Vajpeyi et al. developed a long, thin pressure sensor and applied it to the surface of an endoscope to measure pressure in each part of the colon [8]. Although these studies are focused on bleeding or polyps rather than perforation, or on the development of sensor hardware itself, they provide clues to solve the problems of this study.

In this study, an assisting device is proposed to prevent colonic perforation during colonoscopic insertion. This device can facilitate the procedure, even for novices. Multiple thin and flexible pressure sensors were attached to the colonoscope’s shaft to acquire the pressure signal applied to the colon wall by the shaft during the colonoscopic insertion. These pressure sensor signals were transmitted to the signal processor in real time. Through a series of signal processing, the risk of perforation is categorized into stage 1 (normal), stage 2 (critical), and stage 3 (dangerous) and communicated to the gastroenterologist to recognize an immediate risk and avoid excessive handling of the colonoscope.

Experiments were performed on excised pig intestines to validate the device’s efficacy. First, an expert performed the colonoscopic insertion using this device. The expert determined that the device functioned properly and provided warnings of the risk of perforation. Subsequently, novices carried out the same procedure using the device. As a result, it was found that this device reduced excessive manipulation, thereby lowering the risk of colonic perforation. Therefore, the device can prevent the gastroenterologist from applying excessive pressure to the colon wall. This not only reduces the discomfort of the patient after the procedure but also decreases the risk of perforation.

## 2. Materials and Methods

### 2.1. Pressure Sensing Module

Contact pressure sensors were used to measure the pressure generated on the flexible shaft of the scope during colonoscopy [9]. A small, thin, and flexible sensor was chosen to avoid interfering with the flexible movement of the scope shaft and to minimize patient discomfort. The A101 sensor from Tekscan, Inc. (Boston, MA, USA) was selected, featuring a sensing area of 3.8 mm in diameter, a sensitivity of 1.21 Ω/N, a range of 45 N, and a linearity of 3%. The sensor operates on the principle that when an external force is applied to the sensing area, the internal terminal area deforms downward, increasing the contact area and decreasing the contact resistance, which results in a voltage change. In the device’s circuit, a resistor and the sensor are connected in series. As the force increases on sensing area, the resistance of the sensor decreases, and the voltage applied to the series resistance increases according to Ohm’s law [10]. The pressure is the applied force divided by the sensing area. Therefore, the output voltage from the circuit has a relationship with the applied pressure.

Each pressure sensing module consists of four pressure sensors. The six sensing modules were attached at 10 cm intervals along the scope shaft. The four sensors in each module were arranged at 90-degree intervals along the circumference of the shaft cross-section, as presented in Figure 1. Each sensor was first glued to the shaft surface using a biocompatible adhesive. Each sensor was connected to an external circuit via a very thin, enamel-coated signal wire. The scope shaft with the sensor and signal wires was wrapped and packaged in a latex cover for waterproofing and hygiene. The thickness of the sensor was 0.203 mm, the diameter of the enamel wire was less than 0.2 mm, and the thickness of the latex film was less than 0.1 mm. Therefore, the surrounding of the outer surface of the shaft had little effect on the flexibility or bendability of the colonoscope. The gastroenterologist who handled the proposed device gave a similar opinion. This is also confirmed in a previous study by Vajpeyi et al. [8].

According to the datasheets provided by the manufacturer, the change in the contact resistance with applied pressure is nonlinear. We performed a series of experiments applying different pressures to the sensor and obtained the corresponding output voltage to find out the relationship between the two. The relationship, called the voltage–pressure conversion, was derived through linear regression with a fourth-order linear regression equation, as shown in Figure 2.

### 2.2. Experiments and Data Acquisitions

An intestinal tract phantom was set up for the simulated colonoscopic insertion, as shown in Figure 3. A fresh swine colon was used for the simulation. It was placed on a wooden board and secured using semicircular strap metal clamps to mimic the human large intestine. The overall shape was made U-shaped to replicate the sigmoid and hepatic flexure. Both ends of the intestine were properly closed to prevent the leakage of air or water injected during the experiment.

The data acquisition, signal processing, MCU (Arduino), and fabricated circuit are depicted in Figure 4. Sensors’ voltage signals (*v_i,j_*, *i* = 1, 2, …4, *j* = 1, 2, …6.) were measured from each of the pressure sensors on the colonoscope’s shaft through the circuits. Arduino Mega 2560 (R3) and MUX (Multiplexer) gathered the signal, which was transmitted to MATLAB for signal processing. The processed signals were displayed on an LCD (liquid crystal display) monitor and LEDs, which were mounted right next to the camera controller of the colonoscope. Therefore, physicians could perform simulated colonoscopy using the proposed assisting device on the intestinal tract phantom.

### 2.3. Signal Processing

The output of the proposed device for preventing perforation is derived from the pressure signal applied to the colon wall in real time, and this output should be displayed in a manner that the physician can intuitively recognize. If complex signals are delivered, it becomes difficult for the physician to understand, and if real-time pressure is not reflected, it is challenging to respond immediately. Therefore, it is necessary to calculate the actual pressure signals and process them into a simple perforation warning. Furthermore, a fast response time is crucial to provide real-time perforation warnings during colonoscopic insertion. According to a simple measurement, the time taken for display after applying pressure was less than 0.1 s. There were no problem performing the feasibility tests, but the gastroenterologists suggested that the time should be shorter.

A series of experiments was conducted to deduce the module pressure (*P_j_*) on the colon wall from the sensor signals (*v_i,j_*) and to develop a signal-processing algorithm generating the perforation warning (*E*). Due to the diameters’ difference and the colon’s flexibility, the colon was wound around the shaft, as shown in Figure 5. The module pressure is from one or two sensors when the colon and sensing modules come into contact. A perforation warning was calculated from the module pressure values through the signal processing.

The measured sensor signal (*v_i,j_*) is converted into sensor pressure (*p_i,j_*) through the voltage–pressure conversion for *i*th sensors in the *j*th module (*i* = 1, 2, 3, 4 and *j* = 1, 2, …6). The effect of the fabrication process or atmospheric pressure on the sensors was removed by subtracting the output value before the insertion of the scope into the intestinal tract phantom from the output value after the preliminary insertion. Since the atmospheric pressure is a constant value, it is irrelevant to the role of the device. The pressure from fabrication, such as adhering, wiring, and packaging the sensors to the shaft, also has nothing to do with the device.

For the signal processing, the largest value (*p_max_*_1,*j*_) and the next-largest value (*p_max_*_2*,j*_) were classified among the four pressures for each the pressure sensing modules, as shown in Figure 5. This shows a cross-section of the gray scope shaft and the pink colon wall. The module pressure (*P_j_*) can be determined using the fact that the pressure sensor only detects pressure applied in the vertical direction. If one of the two values is significantly larger, it corresponds to case 1 in Figure 5a. In this case, the largest value (*p_max_*_1,*j*_) is the module pressure (*P_j_*). If the two values are similar, it corresponds to case 2 in Figure 5b. In this case, the module pressure (*P_j_*), is calculated from the following relationship in Equations (1) and (2). This was derived from the fact that the module pressure (*P_j_*) applied at the oblique angle (*θ_j_*) is sensed only in the vertical direction by the two sensors in contact.
(1)$$P_{j}\sin\theta_{j}=p_{max1,j}$$
(2)$$P_{j}\cos\theta_{j}=p_{max2,j}$$

According to a previous study, pressure larger than 200 kPa may lead to hurt or the perforation of the colon wall [11]. Accordingly, the perforation warning (*E*) was set up to have 1, 2, and 3 steps. Step 1 is normal (*P_j_*’s < 100 kPa), step 2 is critical (100 kPa < *P_j_*’s < 150 kPa), and step 3 is dangerous (*P_j_*’s < 150 kPa). The steps were categorized to be easily transmitted to the gastroenterologist through the LED and the display.

The perforation warning allows gastroenterologists to monitor their handling and avoid excessive manipulation. The pressure change is fed back to the physician in real time through the display. The signal-processing method is simple and fast enough to assist the surgeon in real time. During the actual insertion, the gastroenterologist monitors their handling while observing the display with the perforation warning to ensure that excessive pressure is not applied to the colon wall. During the procedure, the module pressure is recorded, allowing the gastroenterologist’s technique to be evaluated.

## 3. Results

### 3.1. Processed Signals

The signal-processing process is depicted in Figure 6. The sensor voltage (*v_i,j_*) was measured for each the pressure sensor, and the voltage data were converted into sensor pressure data (*p_i,j_**) through the voltage–pressure conversion. The effect of the manufacturing process or atmospheric pressure on the sensor was calibrated (*p_i,j_*). Then, the six module pressures (*P_j_*) acting on the colon wall were induced using Equations (1) and (2), as well as Figure 5. The effect of atmospheric pressure was removed by subtracting the average pressure value measured through the sensor in the air and the average pressure value measured after a certain period of time by inserting and stopping the proposed device in the colon phantom. This process also eliminates some of the pressure generated during the fabrication process. This pressure is generated by the residual stress that occurs when assembling sensors, wires, latex, etc. This effect is alleviated after some time by injecting a saline solution similar to body temperature into the phantom.

As a result of signal-processing the pressure caused by the contact between shaft and colon wall, the risk of perforation can be easily distinguished and then delivered to gastroenterologists. All the module pressures are shown in Figure 7, but they are somewhat complicated and difficult for the physicians to understand. To emphasize real-time processing, focusing on the time range from 65 to 70 s, the module pressures were expressed as shown in Figure 8a. They were analyzed again, and the perforation warning was delivered to the gastroenterologists as shown in Figure 8b. If the perforation warning occurs, the gastroenterologist’s corrective action is to stop pushing the shaft or release the loop by looking at the warning on the LED and the signal trend on the monitor mentioned above during the colonoscopy.

### 3.2. Effectiveness Test—A Case Study

To evaluate the effectiveness of the device, physicians and engineers gathered and conducted an experiment, as shown in Table 1. An Olympus endoscope and the proposed device were used as equipment. The insertion target was an intestinal phantom made of intestines extracted from fresh swine carcasses within 8 h of death.

Engineers acquired and processed pressure signals. The processed signals were visually delivered to the physicians via a display device, allowing the gastroenterologist to properly monitor their manipulation. When an excessive pressure signal was detected due to excessive handling, it was fed back to the gastroenterologist through the display, enabling the avoidance of perforation.

The expert gastroenterologist who participated in the experiment, along with the assisting physicians, evaluated that the device was properly made and posed no major inconvenience in use. Both novices and experts may experience excessive pressure when inserting a colonoscope. This aligns with previous studies reporting that even experienced experts may be at risk of perforation [12].

## 4. Discussion

The colonoscope equipped with the proposed device was inserted into the intestinal tract phantom by a novice to validate the device’s effectiveness. As depicted in Figure 9, the module pressure of the device was measured and displayed. When the device was applied, the mean pressure on the colon wall was significantly lower than when the device was not applied. This can be observed by examining the pressure inside the dotted circle. This reduction is attributed to the novice being conscious of the real-time perforation warning. Furthermore, if the excessive pressure exceeded a dangerous level, the novice could monitor the pressure in real time and adjust their manipulation accordingly. Therefore, perforation could be prevented by maintaining low pressure during the insertion procedure.

In Figure 9, a pressure peak at 120 s is shown. This peak occurred because the pressure sensor mounted on the shaft directly collided with the metal clamp holding the intestinal tract phantom. This appears to be an extreme point and is considered to have less relevance to the actual insertion for colonoscopic diagnosis or treatment.

## 5. Conclusions

In this study, we developed an assisting device to prevent colon perforation and ensure safety during colonoscopic insertion for both diagnosis and treatment, even for novices. The pressure data generated on the shaft during the procedure were acquired using pressure sensors. The risk of perforation was then classified through a series of signal-processing steps. The perforation warning could be delivered to the physician through a display that can be intuitively recognized in real time, allowing for the safe performance of diagnosis and treatment with appropriate manipulation.

A series of simulated colonoscopic insertions was conducted on an intestinal tract phantom using a fresh swine colon by physicians and engineers to demonstrate the device’s effectiveness. Both a general colonoscope and the proposed device were applied. The tract phantom was shaped to mimic the sigmoid and hepatic curvature of the human colon. The experts who participated in the experiment evaluated that the device was properly built and posed no significant inconvenience in use.

To demonstrate the device’s effectiveness, the colonoscopic insertion process was tested and compared when a novice used and did not use the device. In this comparison study, it was confirmed that the novice exerted less pressure on the colon wall when using the device. Therefore, if a novice uses the device for colonoscopic insertion, they can lower the average pressure exerted during diagnosis and treatment, thereby reducing the risk of perforation.

Based on this study, expert gastroenterologists can easily perform the necessary techniques for colonoscopy insertion, and even novices can perform diagnosis and treatment with a shorter learning curve. Furthermore, it is possible to alleviate patient suffering and reduce medical costs by lowering the causes of medical disputes. In the future, the design and signal processing of the device should be improved, and more experiments should be performed on live swine models or humans.

For medical devices, long-term environmental stability testing against changes in temperature and humidity is required. Safety and durability are also important concerns. This study is relatively preclinical, so these aspects are not covered faithfully. Despite there being no major problems in the experiment, tests for stability, safety, and durability need to be conducted in future work.

## Figures and Tables

**Figure 1 sensors-24-05711-f001:**
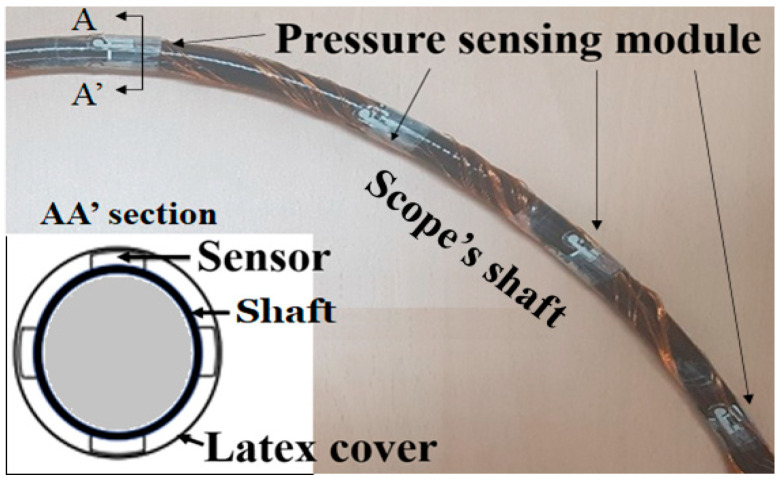
Colonoscopy with pressure sensing modules.

**Figure 2 sensors-24-05711-f002:**
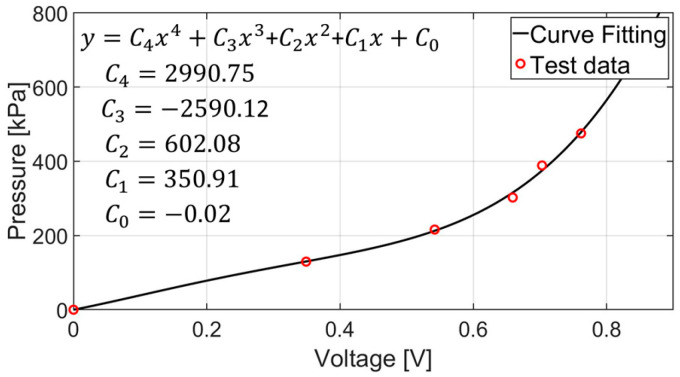
Sensor output voltage vs. applied pressure.

**Figure 3 sensors-24-05711-f003:**
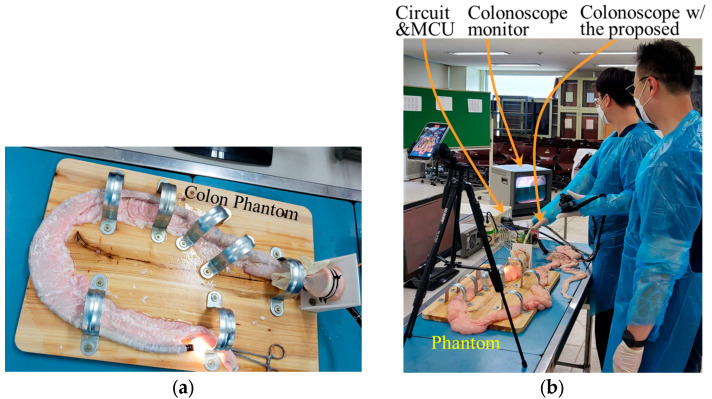
Simulation of colonoscope procedure; (**a**) phantom, (**b**) experiment.

**Figure 4 sensors-24-05711-f004:**
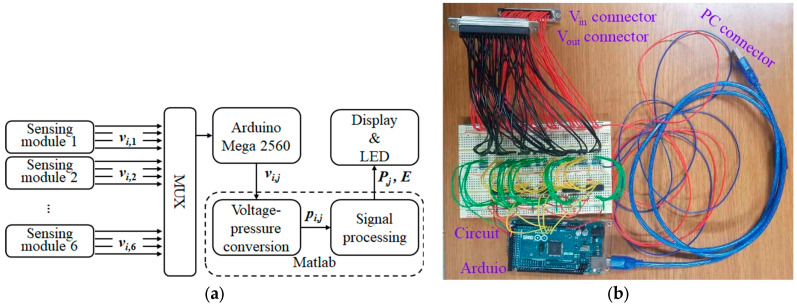
Data acquisition and signal processing. (**a**) Flow of signal and data. (**b**) MCU, circuit and connectors. *v_i,j_* = sensor voltage, *p_i,j_* = sensor pressure, *P_j_* = module pressure (*i* = 1, 2, …4, *j* = 1, 2, …6), *E* = perforation warning (1, 2 or 3).

**Figure 5 sensors-24-05711-f005:**
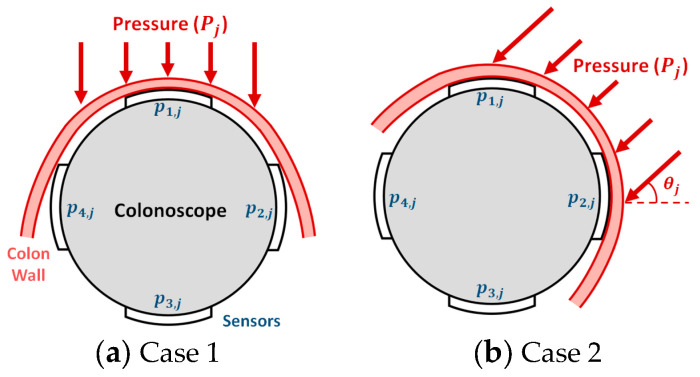
Pressure from colon wall to pressure sensing module.

**Figure 6 sensors-24-05711-f006:**
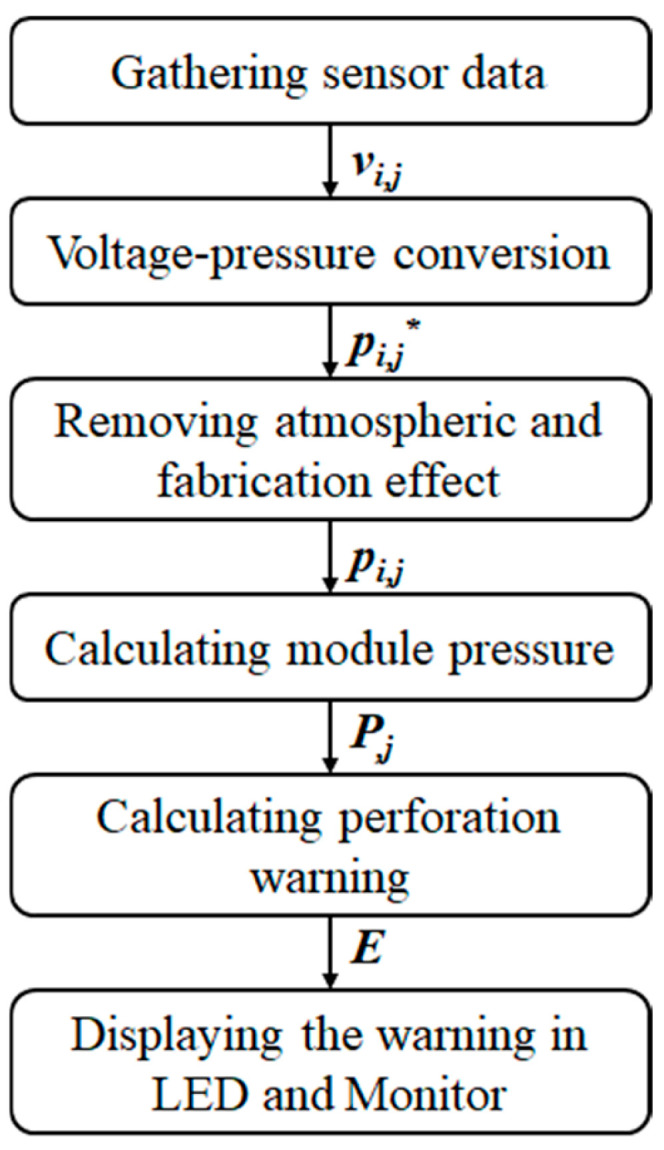
Signal processing; sensor voltage to perforation warning. *p_i,j_** = sensor pressure before removing atmospheric and fabrication effect. (*i* = 1, 2, …4, *j* = 1, 2, …6).

**Figure 7 sensors-24-05711-f007:**
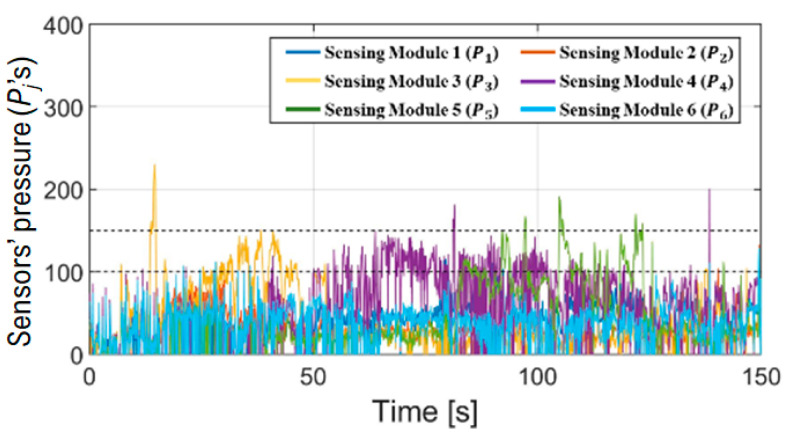
Sensor pressure after voltage–pressure conversion. The gastroenterologist in the study recommended maintaining pressure between the two dashed lines during the procedure to avoid perforation.

**Figure 8 sensors-24-05711-f008:**
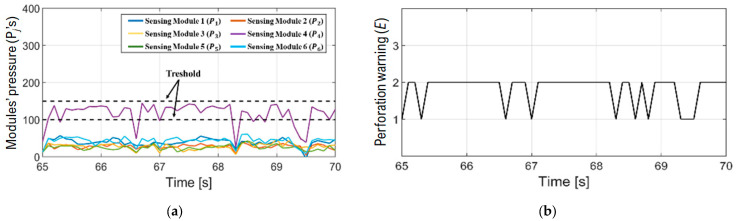
Real-time pressure and perforation warning: (**a**) Module pressure for each module (*P_j_*, *j* = 1, 2, …6); (**b**) Real-time perforation warning (*E* = 1, 2, 3).

**Figure 9 sensors-24-05711-f009:**
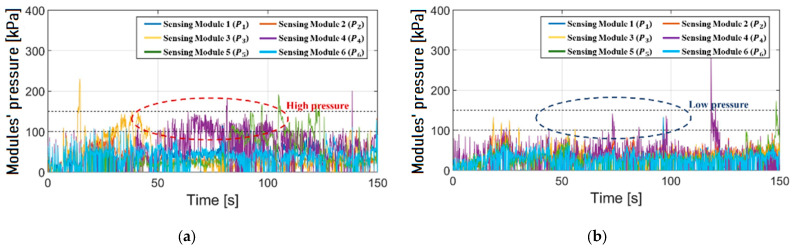
Module pressure of a novice with and without proposed device: (**a**) a novice’s module pressure without the device, (**b**) a novice’s module pressure without the device.

**Table 1 sensors-24-05711-t001:** Experimental overview.

**Date**	28 July 2021 15:00–16:30 (about 90 min)
**Place**	Soonchunhyang University, College of Medicine, Cheonan-si, South Korea
**Participants**	Expert gastroenterologist Novice gastroenterologist Assistant physician General engineer Device engineer × 2 Assistant engineer
**Equipment**	Olympus EVIS LUCERA—ELITE CLV290SL, GIF—HQ290, HX-610-090L
**Target**	Tract phantom made of large intestine from swine carcass (in vitro)

## Data Availability

Data are contained within the article.

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
