# Peer review of "A Pressure Sensing Device to Assist in Colonoscopic Procedures to Prevent Perforation—A Case Study"

_sensors, 2024, doi:10.3390/s24175711_

Round 1
Reviewer 1 Report
Comments and Suggestions for Authors
This manuscript proposes an assistive device that senses the pressure exerted on the colon wall, analyzes the risk of perforation, and warns the physician in real time. The sensor measures pressure signals during colonoscopy. A simple signal processor is used to acquire and process the pressure signal. The processed data is visually transmitted to the gastroenterologist via a display and light emitting diode. The device helps the physician to continuously monitor and prevent excessive pressure on the colon wall. In addition, the device appropriately generates and issues warnings to help the physician prevent colon perforation. The detailed review comments for this review are as follows:
1. In the introduction section there is not enough discussion about wearable sensors to detect intestinal pressure, we note that the literature cited by the authors are research results from several years ago, please add literature from the last two years.
2. Figure 3 shows the principle of data acquisition and signal processing, we suggest the authors to add photos of the acquisition device or related equipment on the right side, and the frame diagram of the authors' Figure 3 is more convincing.
3. The authors need to perform long-term environmental stability tests of the assistive device, such as exposure to different temperatures or humidity, and provide the results.
4. Figure 2 shows a simulated colonoscopy insertion to set up an intestinal prosthesis. Figure 2 in the review PDF file is very blurry, so we suggest that the authors should zoom in on a particular area of the colon where the pressure signal was acquired to highlight the experiment.
5. The authors need to further touch up the language of the manuscript to meet the publication standards of the journal.
Conclusion: Manuscript accepted with major revisions.
Comments on the Quality of English LanguageThe language of the manuscript should be improved.
Author Response
Thank you for carefully reading our paper and providing helpful suggestions and constructive criticism.
Comment 1) In the introduction section there is not enough discussion about wearable sensors to detect intestinal pressure, we note that the literature cited by the authors are research results from several years ago, please add literature from the last two years.
- Answer: The authors thank to the helpful comments. The authors presented the latest biomedical research results as references and explained them as follows; Biomedical engineers have also made efforts to solve problems that arise during en-doscopic diagnosis and treatment. Gerald et al. conducted a study to detect bleeding by attaching a small sensor to the surface of the endoscope shaft. [6] Javazm et al. attached a balloon filled with air to the outside of the endoscope and diagnosed polyps by changes in pressure. [7] Vajpeyi et al. developed a long, thin pressure sensor and applied it to the surface of an endoscope to measure pressure in each part of the colon. [8] Although these studies are focused on bleeding or polyps rather than perforation, or on the development of sensor hardware itself, they provide clues to solving the problems of this study. These contents are in the 4th paragraph of 1. Introduction in red letters.
Comment 2) Figure 3 shows the principle of data acquisition and signal processing, we suggest the authors to add photos of the acquisition device or related equipment on the right side, and the frame diagram of the authors' Figure 3 is more convincing.
-Answer: The authors thank to the helpful comments. The authors added a figure and described some contes as follows; The data acquisition, signal processing, MCU (Arduino), and fabricated circuit are depicted in Fig. 4. … These contents are in the 2nd paragraph of 2.2. Experiments and Data Acquisitions in red letters and Fig. 4(b).
Comment 3) The authors need to perform long-term environmental stability tests of the assistive device, such as exposure to different temperatures or humidity, and provide the results.
- Answer: The authors thank to the constructive opinion. The opinion is essential for medical device development. However, because the proposed device is still under development and this study is a feasibility test, the full testing cannot be performed. The authors explained this limitation as follows; For medical devices, long-term environmental stability testing against changes in temperature and humidity is required. Safety and durability are also an important concern. This study is kind of preclinical, so they are not covered faithfully. Despite there were no major problems in the experiment, the tests for stability, safety, and durability needs to be conducted in future work. This content is in the last paragraph of 5. Conclusions in red letters.
Comment 4) Figure 2 shows a simulated colonoscopy insertion to set up an intestinal prosthesis. Figure 2 in the review PDF file is very blurry, so we suggest that the authors should zoom in on a particular area of the colon where the pressure signal was acquired to highlight the experiment.
- Answer: The authors thank to the helpful comments. Based on the reviewer's opinion, the authors changed the picture to a higher resolution one and placed more emphasis on the experimental setup. These are presented in Fig. 3.
Comment 5) The authors need to further touch up the language of the manuscript to meet the publication standards of the journal.
- Answer: The authors thank to the helpful comments. The authors overall refined the paper to make it easier for readers to read.
Reviewer 2 Report
Comments and Suggestions for Authors
Please check the attachment.

Author Response
Thank you for carefully reading our paper and providing helpful suggestions and constructive criticism.
Comment 1) In the introduction, the latest research related to equipment and application fields needs to be cited to provide a more powerful research investigation.
- Answer: The authors thank to the helpful comments. The authors presented the latest biomedical research results as references and explained them as follows; Biomedical engineers have also made efforts to solve problems that arise during endoscopic diagnosis and treatment. Gerald et al. conducted a study to detect bleeding by attaching a small sensor to the surface of the endoscope shaft. [6] Javazm et al. attached a balloon filled with air to the outside of the endoscope and diagnosed polyps by changes in pressure. [7] Vajpeyi et al. developed a long, thin pressure sensor and applied it to the surface of an endoscope to measure pressure in each part of the colon. [8] Although these studies are focused on bleeding or polyps rather than perforation, or on the development of sensor hardware itself, they provide clues to solving the problems of this study. This content is in the 4th paragraph of 1. Introduction in red letters.
Comment 2) The equations (1) and (2) needs to be mentioned in the text, so readers can understand it more clearly.
- Answer: The authors thank to the helpful comments. The authors mentioned the equations as follows; In the case, the module pressure (Pj), is calculated from the following relationship in Eq. (1) and (2) with the oblique angle (θj). This was derived from the fact that the module pressure (Pj) applied at the oblique angle (θj) is sensed only in the vertical direction by the two sensors in contact. These contents are in the 4th paragraph of 2.3. Signal Processing in purple letters.
Comment 3) The live photos in Figure 2 are not so clear, you might use high resolution photos.
- Answer: The authors thank to the helpful comments. Based on the reviewer's opinion, the authors changed the picture to a higher resolution one. These are presented in Fig. 3(b).
Comment 4) The vertical coordinates of Figure 8a and Figure 8b are not consistent.
- Answer: The authors thank to the helpful comments. Based on the reviewer's opinion, the authors modified the y-axis label to keep the consistency, as shown in Fig. 9(a).
Reviewer 3 Report
Comments and Suggestions for Authors
This article applies pressure sensors to pressure monitoring during colonoscopy, assisting doctors in surgery, and has certain novelty and innovation.
The voltage-pressure conversion equation was derived at line 91. Please supplement the derived conversion equation.
Author Response
Thank you for carefully reading our paper and providing helpful suggestion.
Comment 1) The voltage-pressure conversion equation was derived in the 2nd paragraph of the section 2.1. Please supplement the derived conversion equation.
- Answer: The authors thank to the helpful comments. The relationship between the voltage and the pressure is explained as follows.
… As the force increases on sensing area, the resistance of the sensor decreases, and the voltage applied to the series resistance increases according to Ohm’s law. [10] The pressure is the applied force divided by the sensing area. Therefore, the output voltage from the circuit has a relationship with the applied pressure.…. These are in the 3rd paragraph of 2.1. Pressure Sensing Module in green letters.
… According to the datasheets provided by the manufacturer, the change in the contact resistance with applied pressure is nonlinear. We performed a series of experiments applying different pressures to the sensor and obtained the corresponding output voltage to find out the relationship between the two. The relationship, called the voltage-pressure conversion, was derived through linear regression with a fourth order linear regression equation, as shown in Fig. 2. These are in the last paragraph of 2.1. Pressure Sensing Module in green letters.
Reviewer 4 Report
Comments and Suggestions for Authors
The manuscript reports a pressure monitoring system during colonoscopy to prevent colonic perforations. The design of the system is practical, and one example is given. I recommend the manuscript to be published after addressing the following questions.
1. Provide more details on how the sensors are integrated in the colonoscopy.
2. How will the integration of pressure sensors impact the property of the scope, such as flexibility and bending capabilities?
3. What's the response time for the sensors?
4. Please provide more explanation on how the atmospheric and fabrication effect is removed.
5. If a high pressure is detected, which methods can be taken to reduce the damage?
6. How is the durability of the scope with pressure sensors?
Author Response
Thank you for carefully reading our paper and providing helpful suggestions and constructive criticism.
Comment 1) Provide more details on how the sensors are integrated in the colonoscopy.
- Answer: The authors thank to the helpful comment. They add more detail of the integration and fabrication as follows; Each pressure sensing module consists of four pressure sensors. The six sensing modules were attached at 10 cm intervals along the scope shaft. The four sensors in each module are arranged at 90-degree intervals along the circumference of the shaft cross-section, as presented in Figure 1. Each sensor was first glued to the shaft surface using a biocompatible adhesive. Each sensor is connected to an external circuit via a very thin, enamel-coated signal wire. The scope shaft with the sensor and signal wires is wrapped and packaged in a latex cover for waterproofing and hygiene. These contents are in the 2nd paragraph of 2.1. Pressure Sensing Module in blue letters.
Comment 2) How will the integration of pressure sensors impact the property of the scope, such as flexibility and bending capabilities?
- Answer: The authors thank to the constructive question. The authors were not able to rigorously evaluate how the flexibility and bendability of the endoscope shaft change after applying the proposed device. Nevertheless, it was described as follows from indirect facts; … The thickness of the sensor is 0.203 mm, the diameter of the enamel wire is less than 0.2 mm, and the thickness of the latex film is less than 0.1 mm. Therefore, these surrounding the outer surface of the shaft had little effect on the flexibility or bendability of the colonoscope. The gastroenterologist who handled the proposed device gave the similar opinion. This is also confirmed in a previous study by Vajpeyi et al. [8] This content is in the 2nd paragraph of 2.1. Pressure Sensing Module in blue letters.
Comment 3) What's the response time for the sensors?
- Answer: The authors thank to the constructive question. The authors add an explanation on the response time as follows; … According to a simple measurement, the time taken for display after applying pressure was less than 0.1 s. There were no problem performing the feasibility tests, but the gastroenterologists suggested that the time is to be shorter. This content is in the 1st paragraph of 2.3. Signal Processing in blue letters.
Comment 4) Please provide more explanation on how the atmospheric and fabrication effect is removed.
- Answer: The authors thank to the constructive comment. The calibration process is described as follows; The effect of atmospheric pressure was removed by subtracting the average pressure value measured through the sensor in the air and the average pressure value measured after a certain period of time by inserting and stopping the proposed device in the colon phantom. This process also eliminates some of the pressure generated during the fabrication process. This pressure is generated by the residual stress that occurs when assembling sensors, wires, latex, etc. This effect is alleviated after some time by injecting a saline solution similar to body temperature into the phantom. These contents are in the 1st paragraph of 3.1. Processed Signals in blue letters.
Comment 5) If a high pressure is detected, which methods can be taken to reduce the damage?
- Answer: The authors thank to the good question. However, legally, all the endoscopic diagnosis and treatment must be performed according to the doctor's supervision. If the perforation warning occurred, the gastroenterologist's corrective action is to stop pushing the shaft or release the loop by looking at the warning on the LED and the signal trend on the monitor mentioned above during the colonoscopy. These contents are in the 2nd paragraph of 3.1. Processed Signals in blue letters.
Comment 6) How is the durability of the scope with pressure sensors?
- Answer: The authors thank to the good question. For medical devices, long-term environmental stability testing against changes in temperature and humidity is required. Durability is also an important concern. This study is kind of preclinical, so they are not covered faithfully. Despite there were no major problems in the experiment, the tests for environmental stability and durability needs to be conducted in future work. These contents are in the last paragraph of 5. Conclusions in red letters.
Round 2
Reviewer 1 Report
Comments and Suggestions for Authors Accept in present form
Reviewer 4 Report
Comments and Suggestions for Authors
All the questions are answered in a satisfactory manner.
Comments on the Quality of English LanguageThe quality of the English language is fine.